# Optimized deep learning architectures for the classification of colorectal cancer diagnosis using whole slide images

Salman Mohamed Abdulrahman[1], Mamoon Rashid[1] and Fatima Al-Hashimi[2]

[1] School of Information Communication and Technology, Bahrain Polytechnic, Isa Town, Bahrain
[2] Department of Pathology, Bahrain Defense Force-Royal Medical Services, King Hamad University Hospital, Busaiteen, Bahrain



Corresponding author
Mamoon Rashid,
mamoon.rashid@polytechnic.bh

## ABSTRACT

**Background:** The overwhelming number of cancer cases around the world has expressed a critical need for an automated diagnostic tool to assist pathologists in efficiently handling these cases. Colorectal cancer is one of the most common diseases in the world, increasing yearly. The integration of deep learning architectures in digital pathology has shown promising potential as a supportive tool for assisting pathologists in the diagnosis of cancerous tissues. However, the lack of histopathological image datasets of colon cancer impedes the precise evaluation of deep learning diagnosis techniques.

**Methods:** This study proposes an ensemble model, combining EfficientNetv2 and DenseNet architectures, for the binary classification of colorectal cancer from whole slide images. The framework utilizes a new custom dataset containing histopathological images of colorectal cancer cases divided into benign and malignant classes, collected from Bahrain Defence Force-Royal Medical Services-King Hamad University Hospital in the Kingdom of Bahrain. The dataset comprises a total of 4,694 images, extracted from 227 whole slide images of colorectal cancer patients. However, due to limited computational resources, only 2,000 images were utilized in this study.

**Results:** The proposed model achieved a commendable accuracy of 98%, a perfect precision of 100% and a recall of 96.30%, displaying a high generalization ability and robustness. Furthermore, a comparative analysis was performed, which showed that the proposed model outperformed several state-of-the-art architectures.

## INTRODUCTION

Colorectal cancer (CRC) is one of the most common and widespread types of cancer which ranks second in cancer related deaths worldwide (*Siegel et al., 2023*). It initially begins with the formation of small clumps in the colon called polyps that may turn cancerous over time. CRC is generally divided into two main classes, benign and malignant tumors. Benign tumors are non-cancerous and minimal growth which can be treated easily by surgically removing the tumor. However, malignant tumors are cancerous which tends to

grow rapidly, affecting the neighboring organs (*Hossain et al., 2022*; *Patel, 2020*). In 2022, more than 1.9 million cases of CRC along with 904,000 deaths were estimated globally and are increasing yearly. In the Kingdom of Bahrain, colorectal cancer accounts for 11.8% of all cancer cases, making it the second most common type of cancer in males and females. Furthermore, while analyzing the statistics and patterns, it has been noticed that late-stage diagnosis is frequent which addresses the need for early detection and the introduction of screening (*Bray et al., 2024*; *Kalaji et al., 2024*). Currently, histopathological analysis is the main investigative procedure for cancer diagnosis, where the tumors undergo several phases before placing it in microscopic slides. These phases include slicing the tumor into thin sections, staining the tissue, placing the slices into small cassettes, and processing them through fixation and dehydration steps to prepare them for microscopic examination. These microscopic slides are then provided to expert pathologists in which each slide is examined and analyzed manually for final diagnosis. However, the growing number of cases has proved to be challenging and overwhelming to manage with only a finite number of expert pathologists. Each case has an abundant number of microscopic slides and examining each one manually is time-consuming and inefficient. On the other hand, a microscopic camera can be utilized to capture images of the slides, forming histopathological images for further analysis.

The digital pathology field has seen notable developments with the advancements in artificial intelligence (AI) and deep learning (DL) algorithms, where histopathological images are increasingly being explored for their potential to support diagnostic processes (*Hijazi et al., 2024*). The integration of AI and DL in the digital pathology field has presented promising results in assisting pathologists in managing the growing number of cases. Recent innovations in DL architecture have contributed to advancements in CRC diagnosis research, indicating potential as supportive tools for pathologists in improving diagnostic workflows (*Bousis et al., 2023*). Histopathological images acquire multiple complex patterns and features that are critical for classification. Furthermore, DL architecture has demonstrated strong capabilities in extracting complex and high-level features from images, suggesting their suitability for aiding in the diagnosis of cancerous tissues, though further validation is needed for clinical adoption (*Al-Thelaya et al., 2023*). The complex features and patterns extracted are learned by the algorithm, making it robust and able to classify un-seen images accurately.

The aim of this research is to accurately classify tumors into malignant and benign classes to distinguish cancerous tissues using deep learning architectures. Furthermore, this study intends to investigate the generalization ability and capabilities of deep learning algorithms when subjected to new and un-seen images. New images can lead to discovering new features or patterns that are not included in the existing ones. For real life deployment, health care applications need to be tested in different scenarios and on multiple, diverse datasets to ensure that the deep learning model is generalizing well and classifying accurately. Therefore, testing deep learning algorithms on multiple types of images is essential in proving their reliability in the medical field.

The rest of this article is structured as follows: 'Related Work' provides related works summarizing classification of colorectal cancer. In 'Dataset Acquisition', the formulation

of the dataset is discussed. 'Methodology' discusses the methodology employed in this study. The data analysis is carried out in 'Data Analysis'. The results and their discussion are covered in 'Results and Discussion'. In 'Conclusion and Future Work', we conclude our work and provide future work.

## RELATED WORK

Recently, deep learning algorithms have gained a growing interest in the medical field and underwent numerous experiments on cancer classification. There have been ample prior studies on applying deep learning algorithms to classify colorectal cancer. *Yengec-Tasdemir et al. (2024)* introduced in their study an advanced Supervised Contrastive Learning in combination with Big Transfer model for the early detection of colon adenomatous polyps. They achieved an accuracy of 87.1% on their custom dataset. One limitation stated by the authors is that the complexity of the models employed in this study may deter other researchers from implementing them or building upon their work (*Yengec-Tasdemir et al., 2024*). Prior to this study, *Yengec-Tasdemir et al. (2023)* introduced an advanced method for the early detection of colon adenomatous polyps. Their framework includes an ensemble learning architecture which includes the pre-trained ConvNeXt-tiny and ConvNeXt-base, variants of the convolutional neural network (CNN) family, with stain normalization. Their framework achieved an accuracy of 95% on their custom dataset (*Yengec-Tasdemir et al., 2023*). Following this work, *Sasmal et al. (2024)* proposed a generative adversarial network (GAN) on a semi-supervised framework in their study for colorectal polyp classification using histopathological images. Their approach was performed under two different majority voting schemes, 25% and 50%. Their technique yielded an accuracy of 87.5% and 76.25%, respectively. The study highlights several limitations, including the difficulty of obtaining high-quality and annotated data, where the challenge of capturing complex patterns in histopathological images and the potential loss of global contextual information when using patches of whole-slide images as individual samples, which may reduce the model's classification accuracy (*Sasmal et al., 2024*). *Fu et al. (2024)* proposed a framework that utilizes self-learning sampling for the classification of colon and lung cancer using whole-slide images. They employed ResNet-18 for feature extraction and a self-learning sampling module to select only the relevant features. Their proposed model achieved an accuracy of 89.60% on The Cancer Genome Atlas Lung Squamous Cell Carcinoma (TCGA-LUSC) dataset and 92.50% on the colon cancer dataset. This study's self-learning sampling method has provided new insights and concepts on whole slide images sampling for future studies to build up upon (*Fu et al., 2024*).

Simple CNN models and other more advanced DL models that utilize CNN as a backbone have been the main approach in recent years for cancer classification tasks for their proficient feature extraction capabilities and efficiency. *Kim et al. (2023)* proposed a study to evaluate the efficiency of CNN model in the multi-class classification of colorectal lesions. However, one gap mentioned noted by the authors is the limited number of training images in the dataset which restricts the variety of cases seen by the model. Nevertheless, the model achieved an accuracy of 95.5% on their custom dataset (*Kim et al.,*

*2023*). Furthermore, *Hu et al. (2023)* introduced a new public dataset named Enteroscope Biopsy Histopathological H&E Image Dataset (EBHI) of colorectal cancer images. Their dataset was tested on various CNN-based models including VGG16, Inception-V3, ResNet50. The VGG16 model obtained the best results by achieving an accuracy of 95.37%. This study makes a significant contribution to the field by introducing a new dataset and making it publicly available. The availability of such data is vital, as it supports future research efforts and progresses innovations in digital pathology and artificial intelligence fields (*Hu et al., 2023*). *Xu et al. (2023)* proposed a study that utilizes whole slide images along with attention-based multi-instance learning network and three CNN architectures for the multi-class classification of colon cancer subtypes. Their three-layer CNN model achieved the best performance with accuracy of 83.86%. However, the computational complexity of their work was identified as a challenge in which the authors acknowledged and intended to address in future research efforts (*Xu et al., 2023*). *Riasatian et al. (2021)* proposed an advanced framework where they introduced KimiaNet, a network based on the DenseNet architecture in four configurations. The study also faced some limitations including bias datasets and limited cancer types which may affect the model's generalization ability. The study also faced some limitations including bias datasets and limited cancer types which may affect the model's generalization ability. Nevertheless, the fourth configuration of their model, KimiaNet-IV, achieved an accuracy of 96.80% on the colorectal cancer dataset (*Riasatian et al., 2021*).

Furthermore, several studies addressed the classification of other cancer types from histopathological images using deep learning architectures. *Talib et al. (2024)* proposed two deep learning models, one for segmentation and the other for classification of lung cancer using histopathological images. The pre-processing techniques utilized in this study, which were limited to resizing and horizontal or vertical flipping, were noted to be insufficient to improve the image quality. The model used for binary classification was MinClassNet which is a CNN based model that achieved an accuracy of 98.39% (*Talib et al., 2024*). However, the authors acknowledged that the depth of exploration in their results, especially in identifying the boundaries of cancerous regions, has proven to be downside in their work (*Fu et al., 2023*). Subsequently, *Fu et al. (2023)* proposed a framework that utilizes a deep learning approach for the classification between adenoid cystic carcinoma and basal cell adenoma. Their framework utilizes adaptive feature fusion MobileNet architecture that achieved an accuracy of 97.37%. Other various studies addressed the classification of breast cancer which included deep learning architectures such as ResNet-50, 3-layer CNN, and several others that displayed promising results, reaching 97% (*Ashraf, Alam & Sakib, 2024*; *Rafiq et al., 2023*; *Sajiv et al., 2024*; *Eshun, Bikdash & Islam, 2024*).

This research intends to address several limitations identified in previous studies within the fields of digital pathology and artificial intelligence. These include the dependence on highly complex models to achieve acceptable performance, which often poses challenges for reproducibility and wider adoption. Additionally, earlier work frequently suffers from the use of limited training data, insufficient pre-processing techniques, and a lack of

high-quality datasets. Furthermore, this study targets to bridge these gaps and build upon previous research to advance the existing work in the field.

## DATASET ACQUISITION

While collecting a histopathological dataset, there are several challenges that should be taken into consideration prior to the data collection process. Some challenges may include image size and resolution, classification complexity, variability in staining methods, computational demands, and ensuring consistency across the images within the dataset (*Li et al., 2021*). Histopathological images comprise many qualities and features that can make the images large and of high resolution with a huge number of pixels. This could be a limiting factor that many research groups face as they require significant computational power and resources. Therefore, selecting a suitable number of images to use within the framework is critical to ensure the balance between the computational power and the variability in the dataset used. Furthermore, the accuracy and reliability of the collected images pose a vital challenge as cells in histopathological images tend to overlap and may be structured similarly to the naked eye. Therefore, capturing images from the whole slides that are distinct is essential to maintain diversity in the data. Thus, continuous authentication and verification from an expert pathologist is required to validate the images collected, ensuring reliability within the dataset.

Our dataset was manually assembled by collecting colorectal cancer cases, specifically colonic resections, from 227 whole slide images of colorectal cancer patients that were diagnosed at Bahrain Defence Force-Royal Medical Services-King Hamad University Hospital in the Kingdom of Bahrain between 2015 to 2024 The dataset is available on our git repository which can be accessed by the URL (https://github.com/Salman-Mohamed-ai/DL-CRC.git). Collecting microscopic whole slides from colon resections was the most suitable option for various reasons. Colonic resections comprise multiple subsections including hemi-colectomy, transverse colon-resection, left colon-resection, sigmoidectomy, and rectum-resection, which implies that there are an ample number of specimens to be collected leading to a larger and more diverse dataset. In addition, the primary reason in selecting colonic resections is that the sampling of the specimen includes both normal (benign) tissue and abnormal (malignant) tissue, making it appropriate for the binary classification of colon cancer tissues. Each colonic resection case was processed and placed in microscopic slides for examination. These microscopic slides were collected and then the knowledge of an expert pathologist was used to classify them into benign and malignant classes using a standard light microscope. The expert pathologist did not rely on the original clinical diagnoses of the cases but instead, each slide was evaluated independently as a new case to ensure unbiased and standardized classification based exclusively on histopathological features. After labelling the slides, the Pannoramic MIDI II automatic digital slide scanner by 3D Histech (Budapest, Hungary) was utilized to scan the slide and provide a whole slide image. Then, these scans were used to capture the tissue at 20× magnification level to provide a clear appearance of the benign or malignant components, as shown in Fig. 1.

## Benign Sample Images

Benign Sample 1      Benign Sample 2      Benign Sample 3

## Malignant Sample Images

Malignant Sample 1      Malignant Sample 2      Malignant Sample 3

**Figure 1 Sample images from our custom dataset.**

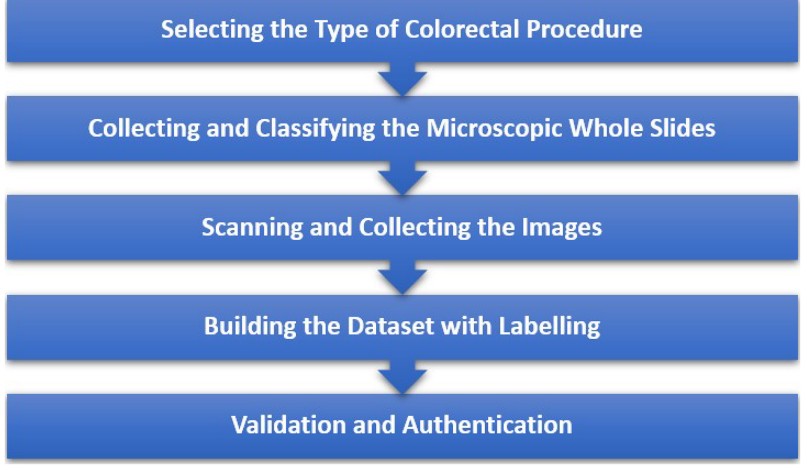

**Figure 2 Steps taken to ensure precise data collection.**

Our thorough and systematic approach to building this custom dataset ensures a precise collection and labelling of the images, which is critical to maximize the diagnostic accuracy of our proposed framework. Furthermore, Fig. 2 illustrates the entire data collection process that provides an overview of the steps taken to ensure accurate data collection.

Upon completing the process, the finalized custom dataset contains a total of 4,694 image patches of the microscopic slides, divided equally into their respective classes. However, this research utilizes only 2,000 images from the dataset due to the lack of suitable computational resources. The number of images used in each class for the training and testing sets is provided in Table 1.

## METHODOLOGY

This section outlines the systemic approach taken to develop a deep learning pipeline for the binary classification of colorectal cancer for the dataset assembled by collecting

**Table 1 Number of images used in each class.**

| Class | Training | Testing |
| --- | --- | --- |
| Benign | 700 | 300 |
| Malignant | 700 | 300 |
| Total | 1,400 | 600 |

colorectal cancer cases, specifically colonic resections, from patients that were diagnosed at Bahrain Defence Force-Royal Medical Services-King Hamad University Hospital in the Kingdom of Bahrain. All the experimentations and trials are implemented using an ASUS laptop with an Intel(R) Core (TM) i5-8265U CPU chip, 8 GB RAM, 512 GB SSD, and a 64-bit operating system along with x64-based processor. The environment utilized for this implementation is Google Colaboratory (version: 1.13.5). This work was carried out by following all the ethical guidelines of the Bahrain Defence Force-Royal Medical Services-King Hamad University Hospital in the Kingdom of Bahrain and was approved by the Institutional Review Board (IRB) under the approval number RMS-KHUH/IRB/2024-810. It includes various pre-processing steps prior to training the model and the deployment of pre-trained models provided in TensorFlow library.

## Data pre-processing

Several pre-processing steps are applied to the dataset to enhance the overall quality, consistency, and variety of the images which in return improves the performance of the model. Initially, all the images were rescaled to normalize the pixel values to a common scale, leading to a faster convergence during training. In addition, all the images were resized to (224,224), to match the expected input dimensions of the model and add consistency in training. Furthermore, to reduce the issue of overfitting, augmentation techniques were applied on the images to add variety to the dataset and improve the model's robustness. The images were subjected to random rotation, shifting, shearing, zooming, and horizontal flipping up to a certain threshold which results in a diverse dataset while maintaining the important features of the images.

Initially, the training set underwent image rescaling and resizing to normalize pixel values to a uniform range and convert the image to a standard size. In addition, data augmentation is implemented to diversify our dataset by applying random transformations such as rotation, shifting, zooming, shearing, horizontal flipping, and brightness adjustment to enhance the model's ability to generalize to raw un-seen images. The rotation augmentation technique was implemented to rotate images randomly up to 30° to introduce angular variations in the images, allowing the model to recognize objects or patterns in various orientations. Similarly, shifting is applied to randomly shift the images horizontally or vertically up to 30% to simulate slight movements in the camera position, strengthening the model's accuracy when subjected to image features at different positions. Furthermore, shearing is employed randomly with a maximum of 30% to skew the image, adding variety in the form of distortion that can help the model learn to recognize shapes even when distorted.

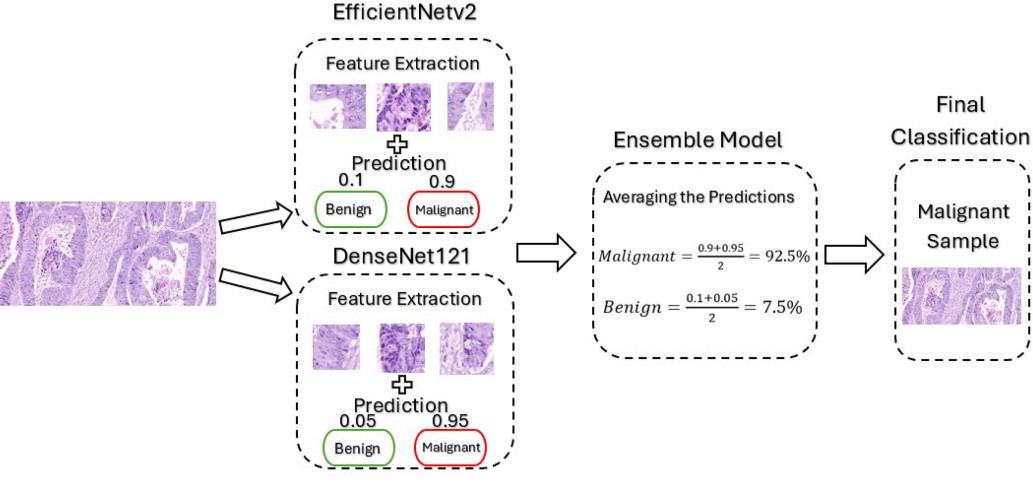

**Figure 3 Architecture of our proposed model.**

## Proposed model architecture

The proposed framework mainly utilizes pre-trained CNN-based models, namely EfficientNetv2B0 and DenseNet121. The EfficientNetV2 model was selected for this framework due to its ability to balance performance with computational efficiency, which is critical given the limited computational resources available for this project. Its architecture incorporates advanced components like MBConv and Fused-MBConv that optimizes both training speed and feature extraction. This makes the model particularly well-suited for binary classification of medical images, which often have high resolution and complex features. Furthermore, DenseNet offers several advantages for cancer classification, particularly its dense connectivity and feed-forward structure, which enhances feature extraction and parameter efficiency. This design helps prevent overfitting while improving generalization, leading to better performance. Despite its deep architecture, DenseNet is computationally efficient, making it well-suited for high-resolution medical images where computational resources are limited. Its ability to handle complex features makes it ideal for binary cancer classification, ensuring a reliable and well-converged model. These models are trained separately and then combined to create an ensemble model where their predictions are averaged for final classification. By creating an ensemble model, we improved the accuracy of the framework by combining the strengths of both models, reducing overfitting by averaging the predictions, and enhancing the overall generalization ability by utilizing the feature extraction techniques employed by the advance models. Figure 3 illustrates the architecture of the model when it is subjected to an un-seen malignant image from our dataset for classification.

## Feature extraction

Histopathological images acquire plenty of features and patterns that are significant for classification. The extraction of these high-level features is critical to accurately classify

these images. The architectures employed in this proposed framework comprises of advanced feature extraction components, making them suitable for this application.

## EfficientNetv2

The EfficientNetv2 contains three main components for high-level feature extraction. First, the image is processed in an initial convolutional layer to extract basic features from the image such as edges and textures. Following this, the fused-mobile inverted bottleneck convolutional layers are used to extract low to mid-level features from the input image, which then combines these features to prepare them for a more complex feature extraction in the later stage. Finally, mobile inverted bottleneck convolutional layers are used in the deeper stages of the model as it is suitable in handling small feature maps and extracting mid to complex or high-level features from the images. This advance feature extraction process enables the model effectively to learn the relevant features and accurately classify them (*Tan & Le, 2021*).

## DenseNet121

DenseNet structure includes an initial convolution and pooling layer followed by four dense blocks with bottleneck layers, and transition layers between the dense blocks for complex feature extraction. Similarly, the initial convolution layer is used to extract low-level features and reduce the spatial dimensions of the image. Then, the four dense blocks perform a more complex operation where their structure is designed in a feed-forward manner to extract high-level features and pass its output to the next layer for further use. The structure of a single dense block includes various identical-interconnected layers, where each layer is designed to focus primarily on feature extraction. This densely structured architecture is beneficial for feature extraction as it allows feature reuse while extracting complex patterns.

## Model training

In order to ensure suitable training procedure of the models, part of the training images was reserved for validation. The validation strategy used in this research was hold-out validation, where 20% of the training data was reserved for validation, within the ImageDataGenerator. To prevent overfitting, two key regularization techniques were applied, L2 regularization on the final dense layer and dropout with a rate of 0.5 before the output layer. During our experimentation, the model underwent several hyperparameter tuning and optimization phases to obtain the best performance possible. These hyperparameters are critical to the training process and the development of the model. Choosing the optimal set of hyperparameters improves accuracy and generalization along with enhancing the robustness of the model. The batch size refers to the number of training samples that pass through the network at one iteration. A small batch size can lead to better convergence while a larger one leads to a faster training process so selecting the ideal batch size is essential to balance between the two. For the proposed model we opted for a batch size of 32 after several trials. Furthermore, the learning rate controls the step size at each iteration and a high value can cause the model to converge quickly while a low

value can lead to getting stuck at a local minimum. After experimentation we set the learning rate to be 0.00003 initially and employed a learning rate scheduler to adjust the value if there is no improvement within five epochs. The epochs describe the number of complete passes through the training set. The balance between high and low number of passes is essential to avoid overfitting or underfitting of the model. The proposed model was trained with 20 epochs which proved to be sufficient for the model to train without overfitting. In addition, early stopping was employed to stop the training process if there are no improvements within five epochs to further avoid overfitting. Finally, Adam optimizer was chosen as its properties proved to be suitable for the model and its application.

## DATA ANALYSIS

In this section, a comprehensive analysis will be performed to compare the original image with the augmented one, signifying the impact of the pre-processing techniques applied in the framework. Data analysis serves a crucial role in understanding and analyzing the characteristics and features of the dataset used for training the proposed colon cancer classification model. The analysis covers a series of comparison techniques including a visual comparison, histogram evaluation, statistical metrics, and feature extraction techniques to demonstrate the impact and benefit of the pre-processing and augmentation techniques implemented in the network.

### Visual comparison

A sample image was used for the visual comparison, from the benign category to demonstrate the impact of the techniques implemented. In addition, the sample underwent the pre-processing and augmentation steps five separate times, where it was altered randomly to demonstrate the variability that these techniques offer. The visual comparisons shown in Fig. 4 illustrate the variations introduced by the augmentation techniques, making the dataset more diverse while still retaining the essential histological features and patterns. Adding diversity to the dataset offers key benefits such as preventing overfitting and improved generalization. These techniques proved to alter the dataset in a way where it balances the variability and the preservation of essential features in the images, leading to a model with increased robustness and superior performance.

### Histogram pixel analysis

In histopathological images, the color intensity of the image and its contrast is essential in accurately diagnosing the tissue. Preserving these color intensities in the image is essential to ensure that the model is trained on realistic and well-defined images. Histogram analysis will be utilized to plot the pixel intensities of each color channel of the image before and after augmentation to show the alterations in contrast and brightness and to ensure that while the augmentation adds variety, there are no major alterations in the colors of the image. A malignant sample was used to compare the color distribution before and after pre-processing. The plots shown in Fig. 5 display that the pixel intensity distribution between the original and the augmented images are relatively similar, indicating that the

Benign Sample

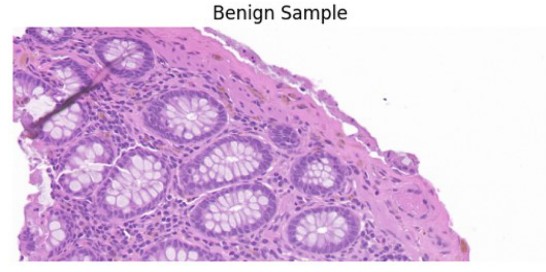

Augmented Versions of the Benign Sample

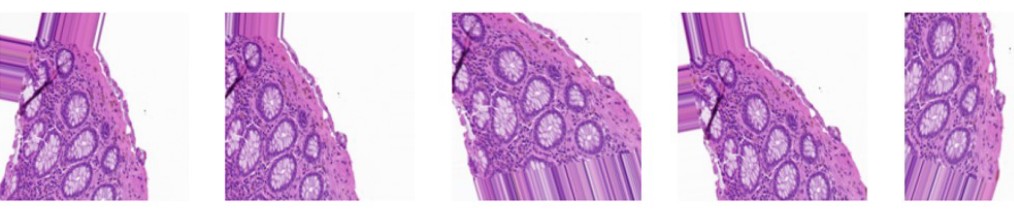

**Figure 4 A benign sample which is augmented five separate times.**

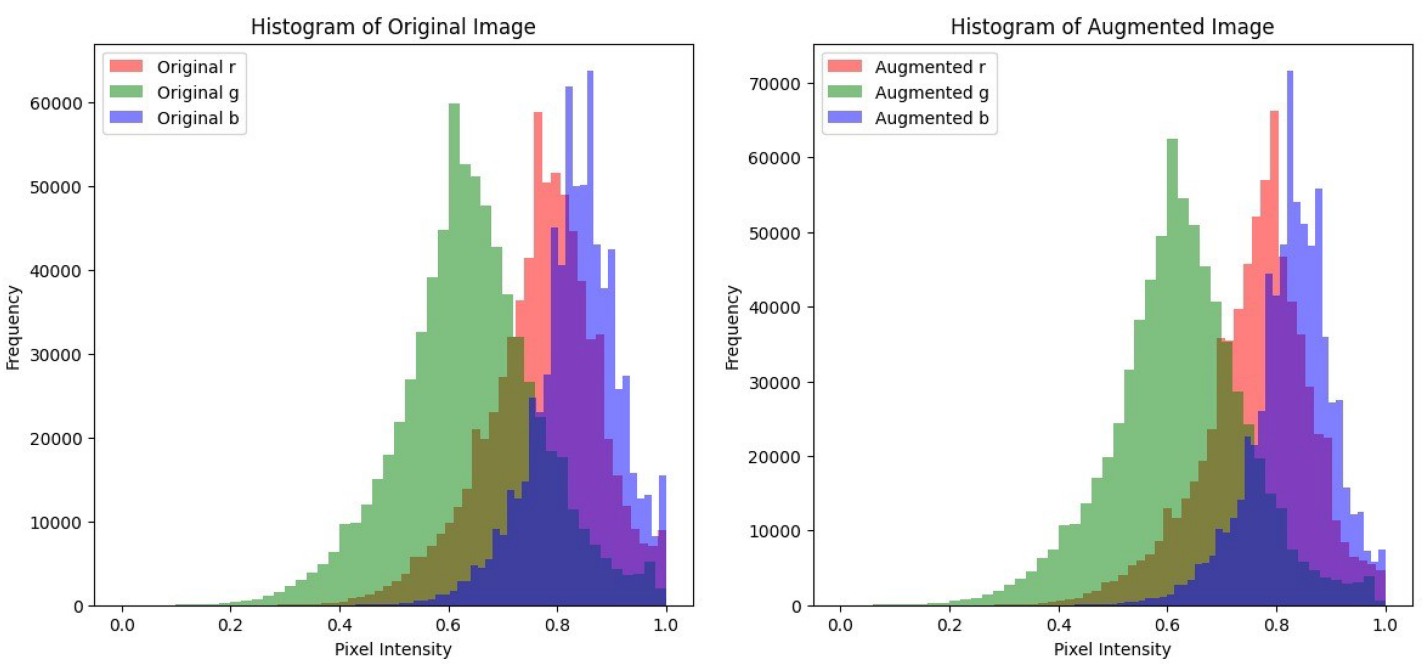

**Figure 5 Histogram plots to compare pixel intensity distributions of the original image and its augmented counterparts.**

overall visual appearance and color distribution remains consistent and is unaffected despite the alterations performed during pre-processing. This consistency in pixel intensities is vital to ensure that textures and patterns in histopathological images remain undisturbed and thus enhance the model's classification accuracy.

## Statistical analysis

Statistical metrics are utilized to analyze the consistency and feature preservation after the pre-processing stage. Metrics such as mean, standard deviation, structural similarity index (SSIM) are employed on the image to perform this statistical analysis.

### Mean

The mean provides an average of the pixel intensities in the image to measure the overall brightness in the original and augmented image. The mathematical formula to obtain the mean value is presented in Eq. (1).

$$\frac{1}{N} = \sum_{i=1}^{N} x_i z \; (intensity \; value \; of \; the \; specific \; pixel). \tag{1}$$

The original image resulted in a mean value of 0.74845 and the augmented image resulted in a mean of 0.74234. The mean values of the original and augmented image are comparatively similar, indicating that the brightness of the image is preserved despite the alterations implemented on the image.

### Standard deviation

The standard deviation metric provides a measure of the spread or distribution of pixel intensities around the mean of an image, representing the contrast of the image. The mathematical formula to calculate the standard deviation is presented in Eq. (2).

$$std = \sqrt{\frac{1}{N} \sum_{i=1}^{N} (x_i - Mean)^2}. \tag{2}$$

The original image resulted in a standard deviation value of 0.13302 and the augmented image obtained a value of 0.13233. The standard deviation values from the original and augmented image are almost similar, indicating that the augmentation techniques implemented on the image preserved its contrast.

### Structural similarity index

SSIM is a perceptual metric used for the comparison of the structural differences between the original and augmented image. Its values range from −1 to 1, with 1 indicating a perfect similarity in structure while 0 suggests no similarity and −1 signifies that the images are completely different with no correlation, calculated as presented in Eq. (3). SSIM shows the impact made by the augmentation techniques on the images. If the resulted value is less than 0, it signifies that adjustments and tuning are required on the augmentation stage before training.

$$SSIM(x, y) = \frac{\left(2\mu_x\mu_y + C_1\right)\left(2\sigma_{xy} + C_2\right)}{\left(\mu_x^2 + \mu_y^2 + C_1\right)\left(\sigma_x^2\sigma_y^2 + C_2\right)}. \tag{3}$$

The SSIM value obtained while comparing both the original and the augmented image was 0.3262, which indicates a moderate similarity in the structure of the images. The augmentation techniques implemented on the image such as rotation, shifting, zooming

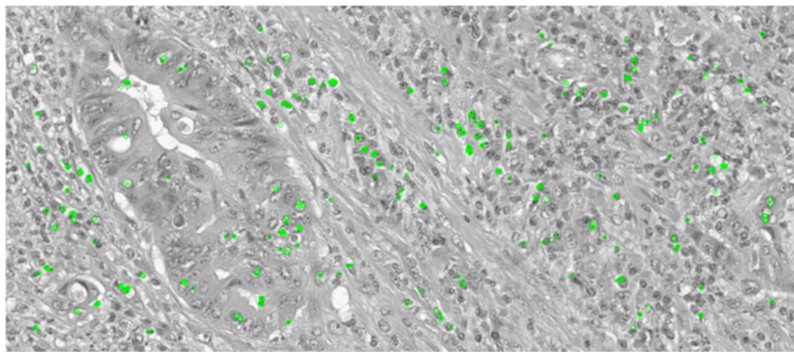

**Figure 6  Keypoint detection applied on the original image.**

and horizontal flipping, added variability to the dataset but altered the structure of the image significantly. Therefore, SSIM values decreased slightly but nevertheless, still showed moderate similarity between the structures of the images. In addition, other metrics are applied to ensure that the essential features of the images are well preserved and not majorly effected by the augmentation techniques applied.

## Keypoint detection

Keypoint detection is essential in this comparative analysis as it will determine if the essential and key features of the images are preserved or distorted after the augmentation techniques are applied. In histopathological images, each category encompasses many key features, distinguishing them from other classes. Therefore, preserving these essential features is crucial to ensure accurate classification of colon cancer tissues. Oriented FAST and Rotated BRIEF (ORB) is a feature detector and descriptor extractor model which is utilized to inspect the original and augmented images, detecting features to ensure that the key features are preserved and not effected by the augmentation. The key point detection was applied on an original image of a malignant sample and then on an augmented version of the same sample where it was rotated, shifted, flipped, and zoomed to demonstrate if the keypoint detection can extract similar key features from the image.

The ORB keypoint detection, shown in Figs. 6 and 7, established that the key features in these histopathological images remain consistent after augmentation as it was able to retain majority of these features of the image even after it was subjected to the augmentation techniques. This implementation verified that the essential features are preserved as they are critical for maintaining the diagnostic accuracy and value of the histopathological images.

This comparative analysis demonstrated that the pre-processing steps and augmentation techniques applied on the dataset enriched the diversity of the images while preserving the essential and important features that are vital for accurate classification and diagnosis. This multi-faced analysis authenticates the pre-processing methodology implemented and highlights its importance for efficient training of the models, preceding to a reliable and superior classification performance.

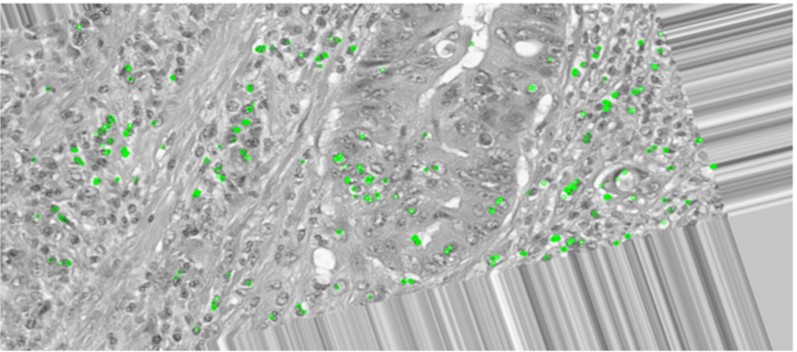

**Figure 7 Keypoint detection applied on the augmented image.**

## Assessment metrics

The performance metrics utilized to evaluate the framework are the following:

### Confusion matrix

A confusion matrix is a table mainly used to evaluate a classification model by providing an overall count of the true positive (TP), true negative (TN), false positive (FP), and false negative (FN) predictions.

### Accuracy

The accuracy metric gives an insight into the percentage of the correctly predicted cases among the total number of cases, as presented in Eq. (4).

$$Accuracy = \frac{TP + TN}{TP + TN + FP + FN}. \tag{4}$$

### Precision

Precision provides an insight into the model's accuracy and reliability in predicting positive cases. Equation (5) shows us how precision value is obtained.

$$Precision = \frac{True\ Positive}{True\ Positive + False\ Positive}. \tag{5}$$

### Recall

Recall provides an insight into the model's ability to find all positive cases, which is essential for medical applications. Equation (6) shows us how the recall value is obtained.

$$Recall = \frac{True\ Positive}{True\ Positive + False\ Negative}. \tag{6}$$

## RESULTS AND DISCUSSION

In this section, we will display the results obtained from the proposed ensemble framework and discuss them along with comparing our framework with the state-of-the-art.

| Table 2 Individual performance of the models on the training set. | | |
|---|---|---|
| Metric | EfficientNetv2B0 | DenseNet121 |
| Training accuracy | 99.37% | 99.82% |
| Validation accuracy | 99.29% | 98.57% |
| Training loss | 0.0270 | 0.0079 |
| Validation loss | 0.0352 | 0.0452 |

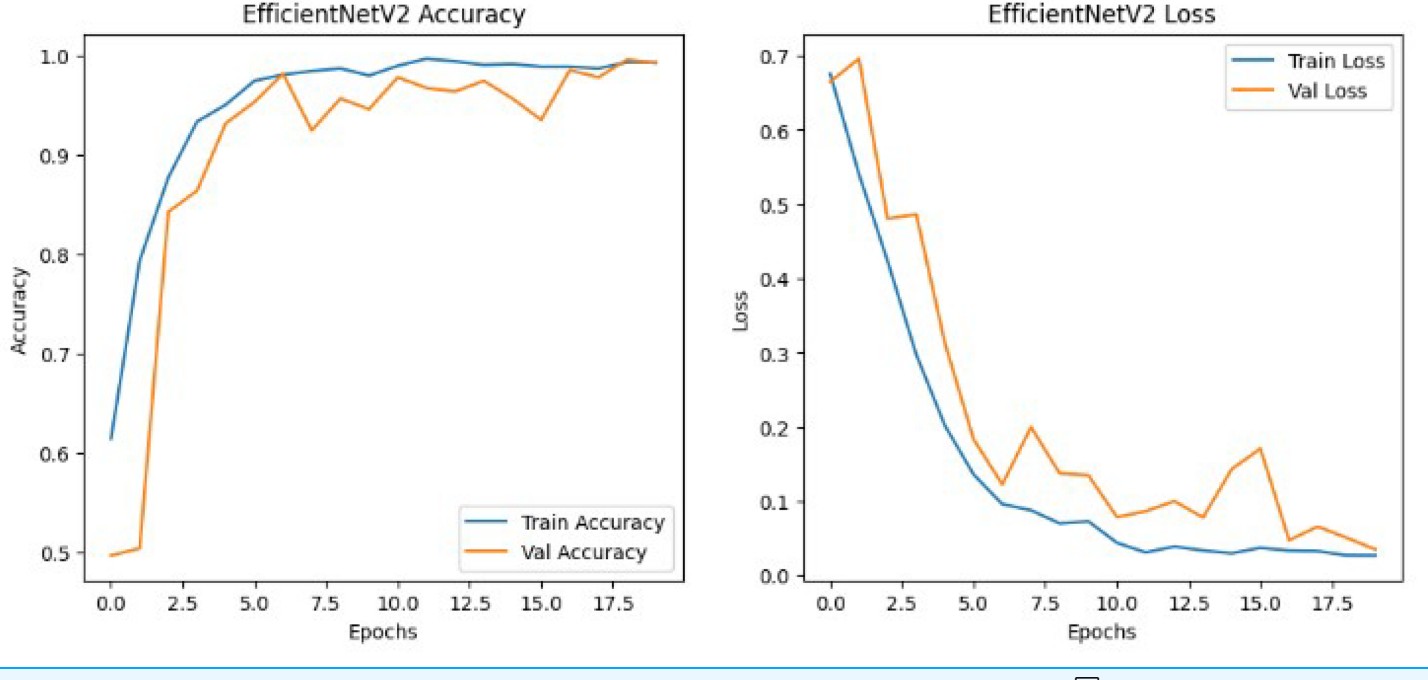

Figure 8 EfficientNetv2 training history.

In this work, out of 1,400 images, we utilized 1,120 images for training and kept 280 images for validation to monitor the individual model's performance during the training phase. Table 2 compares the results achieved by the models in training and validation.

The results shown in Table 2 show that the models perform similarly on the dataset with EfficientNetv2B0 having a slightly better performance than the DenseNet121 architecture in the validation set. Although EfficientNetv2B0 had better results in the validation set, the training history, visualized in Figs. 8 and 9, showed that DenseNet121 was more well-fitted to the data as it did not exhibit any indication of overfitting or noise during training.

The proposed ensemble model was evaluated on the test set, which consisted of 600 images, to evaluate its performance and generalization ability to un-seen images. The test set was equally divided between the classes to perform a fair evaluation of the model. Table 3 displays the performance of the ensemble model on the test set along with the performance of several other state-of-the-art methodologies. The results achieved by the ensemble model proved that by combining the EfficientNetv2B0 and DenseNet121 architectures to create an

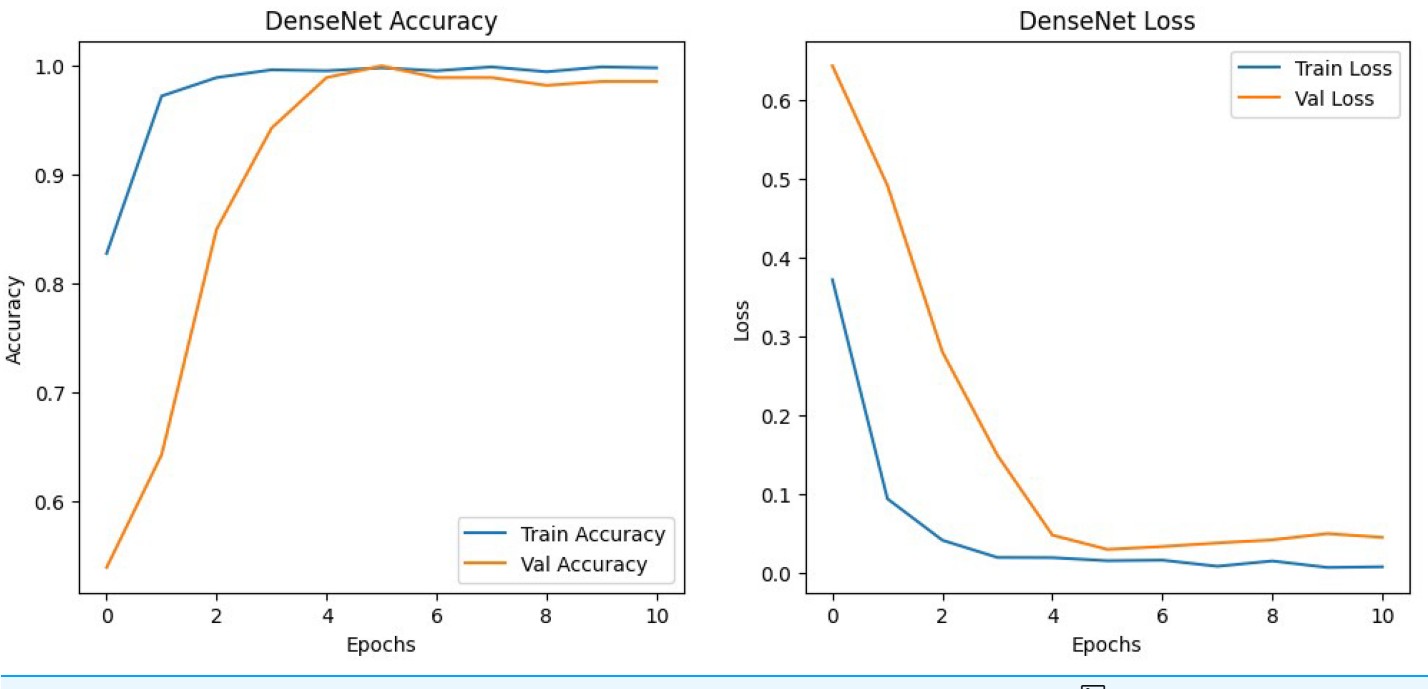

**Figure 9 DenseNet training history.**

**Table 3 Performance of proposed model with the state-of-the-art.**

| Metric | Proposed model | Yengec-Tasdemir et al. (2024) | Sasmal et al. (2024) | Fu et al. (2024) | Ashraf, Alam & Sakib (2024) | Yengec-Tasdemir et al. (2023) | Fu et al. (2023) |
|---|---|---|---|---|---|---|---|
| Testing accuracy | 98% | 87.1% | 87.5% | 92.5% | 98% | 95% | 97.8% |
| Precision | 100% | 86.3% | – | – | – | 92.8% | – |
| Recall | 96.30% | 86.2% | – | – | – | 95.1% | 97.3% |

ensemble model, it presented an excellent performance. The proposed model achieved admirable accuracy, perfect precision, and a high recall along with the perfect classification of the benign samples, showcasing the model's reliability, robustness, and effectiveness in accurately classifying histopathological images. The 98% accuracy achieved by the model on the test set proved that the proposed model could generalize well when subjected to un-seen images. Correspondingly, the model yielded a perfect precision of 100% which insinuates that every sample classified and predicted as a malignant sample is indeed malignant and the model is extremely reliable when it classifies a sample as malignant. Additionally, the model achieved a high recall of 96.30%, which implies that the model accurately classified most of the malignant samples and has a very low false negative rate. Furthermore, the confusion matrix, shown in Fig. 10, demonstrated that the model perfectly classified all the benign samples and only misclassified 3% of the malignant samples. These misclassifications are likely due to subtle patterns that are difficult even for human pathologists to discern. The values obtained from the actual and predicted sets in the confusion matrix indicates that the model can classify the images nearly perfectly into the

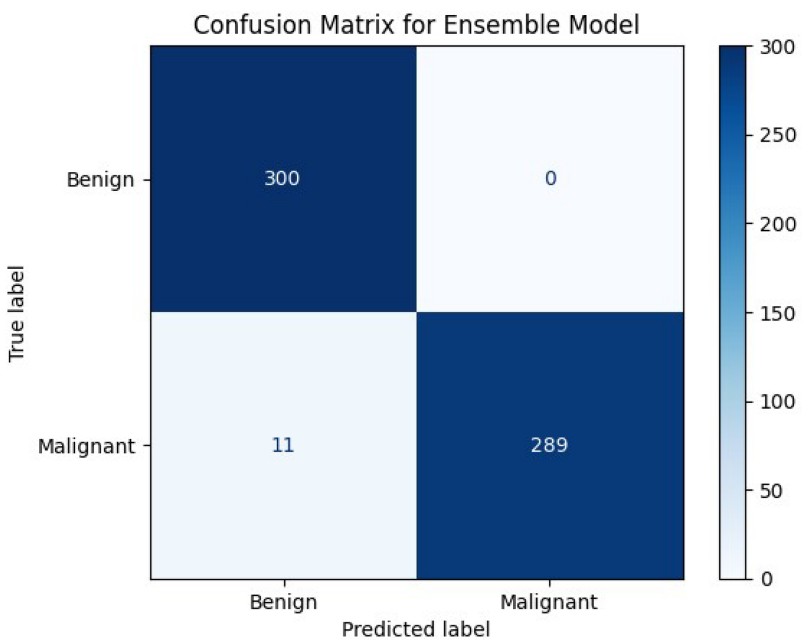

**Figure 10 Confusion matrix of our proposed model.**

appropriate category, highlighting the model's distinguishing power and the ability to accurately classify between the classes. The inclusive inference is that the proposed model exhibited excellent performance and a strong generalization ability.

The performance achieved the expected outcome, leaving slight room for future improvement and adjustments to further enhance the model's overall performance. Furthermore, the proposed model outperformed or is aligned with various recent state-of-the-art methodologies. The comparison showed that the proposed model significantly outperformed them in terms of accuracy. It is important to note that the comparison presented in Table 3 may not be entirely equitable, as the proposed model was trained and tested on un-seen data from the same dataset it is trained with, while the other referenced models were evaluated on datasets from different locations or distributions. In addition, some of the referenced models did not report precision and recall metrics, and therefore these values were not included in Table 3. However, this comparison still provides a useful benchmark, offering insights into how the proposed approach performs relative to existing methods under similar evaluation settings, and highlights its potential when applied to data from the same clinical context. Nevertheless, by combining the predictive power of these advanced architectures along with the pre-processing steps and augmentation techniques has proven to be an effective and efficient approach for colon cancer classification, positioning it as a competitive framework within the recent state-of-the-art research.

## CONCLUSION AND FUTURE WORK

In this article, we introduced a new custom dataset encompassing histopathological images of colorectal cancer divided into benign and malignant categories and an ensemble model that leverages transfer learning by combining two pre-trained models, EfficientNetV2 and

DenseNet, for the binary classification of colon cancer using the custom dataset. The ensemble model exhibited a remarkable performance and robustness in extracting high-level features from the dataset and accurately classifying them. Furthermore, a comprehensive comparative analysis was conducted, demonstrating that the proposed model consistently outperformed several state-of-the-art architectures in terms of accuracy, robustness, and overall performance across multiple evaluation metrics. This work is an initial attempt to acquire and utilize histopathological data of colorectal cancer patients that were diagnosed at Bahrain Defence Force-Royal Medical Services-King Hamad University Hospital in the Kingdom of Bahrain. The primary goal is to develop a solution tailored to the needs of Bahrain Defence Force-Royal Medical Services-King Hamad University Hospital, ensuring that the institution benefits directly from the outcomes of this research through improved diagnostic support and clinical decision-making.

Nonetheless, there are several downsides and limitations acknowledged in this study, which can be improved in the future. The custom dataset can be expanded to enhance the model's training and subject it to more diverse images and improve the robustness and accuracy of the proposed model. In addition, the model is trained and dependent on only one dataset which may limit the model's generalization ability and make it biased to only these types of images even after augmentation. In light of this, further validation on other datasets may be required to confirm the model's performance. Furthermore, the proposed framework focuses mainly on histopathological images related to colorectal cancer and no other cancer types. This class specificity limits the model to only one cancer class and potentially makes it unreliable when tested on other classes. The work done has built a solid foundation and proposes various opportunities for future work and enhancements to further enhance the model and address the limitations experienced during the study.

### Funding
This research was supported for publication by Bahrain Polytechnic's institutional funding with Ref. number AREC/PUB/2025/022. The funders had no role in study design, data collection and analysis, decision to publish, or preparation of the manuscript.

### Grant Disclosures
The following grant information was disclosed by the authors:
Bahrain Polytechnic's Institutional funding: AREC/PUB/2025/022.

### Competing Interests
The authors declare that they have no competing interests.

### Author Contributions
- Salman Mohamed Abdulrahman conceived and designed the experiments, performed the experiments, performed the computation work, prepared figures and/or tables, and approved the final draft.

- Mamoon Rashid conceived and designed the experiments, analyzed the data, authored or reviewed drafts of the article, and approved the final draft.
- Fatima Al-Hashimi analyzed the data, performed the computation work, prepared figures and/or tables, and approved the final draft.

### Ethics

The following information was supplied relating to ethical approvals (*i.e.*, approving body and any reference numbers):

The study was approved by the Institutional Review Board (IRB) at Bahrain Defense Force-Royal Medical Services-King Hamad University Hospital, Kingdom of Bahrain, approval number: RMS-KHUH/IRB/2024-810.

### Data Availability

The DL-CRC data is available at GitHub and Zenodo:

- https://github.com/Salman-Mohamed-ai/DL-CRC.

- Salman-Mohamed-ai. (2025). Salman-Mohamed-ai/DL-CRC: First Release (V1.0). Zenodo. https://doi.org/10.5281/zenodo.17104874.

### Supplemental Information

Supplemental information for this article can be found online at http://dx.doi.org/10.7717/peerj-cs.3241#supplemental-information.

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
