# Peer review of "Optimized deep learning architectures for the classification of colorectal cancer diagnosis using whole slide images"

_PeerJ Computer Science, doi:10.7717/peerj-cs.3241_

## Round 0.1 · original submission · Minor Revisions

· Academic Editor

Minor Revisions

**Language Note:** The review process has identified that the English language must be improved. PeerJ can provide language editing services - please contact us at [email protected] for pricing (be sure to provide your manuscript number and title). Alternatively, you should make your own arrangements to improve the language quality and provide details in your response letter. – PeerJ Staff

·

Basic reporting

1. While the authors summarize previous work, adding a brief critical analysis of each study's limitations or contributions would add more depth. This would also help to position the current study's novelty and contribution more effectively.
2. The paper should explicitly identify the gaps in research that the current study aims to fill. What specific limitations of previous studies does this research address?

Experimental design

1. The description of the ensemble model is clear. However, providing a visual representation of the architecture (e.g., a diagram) would be very helpful for the reader. The explanation of EfficientNetv2 and DenseNet121 is good but consider adding a sentence or two about why these specific architectures were chosen.
2. The training details are sufficient. However, consider adding details about the validation strategy (e.g., k-fold cross-validation) and any regularization techniques used.

Validity of the findings

1. The results are clearly presented, and the tables and figures are generally effective. However, ensure that all table and figure captions are self-explanatory.
2. The performance metrics are impressive, consider adding statistical significance testing to support the claim that the proposed model "significantly outperformed" other state-of-the-art methods if that's the case.
3. A more in-depth analysis of the model's errors. What types of images were misclassified? Why might this have happened?

Additional comments

1. The paper is generally well-written, but a thorough proofreading is recommended to eliminate any minor grammatical errors or typos.
2. Include a more detailed discussion of the limitations of the study and potential sources of bias.

Reviewer 2 ·

Basic reporting

This manuscript describes the use of a deep learning-based ensemble model for classifying whole slide images from colorectal resections into malignant or benign. In general, the manuscript provides a clear, coherent description of the authors’ approach and findings. However, some essential details are lacking in several places which would help improve clarity and aid interpretation/replication. This includes the following:
- Abstract:
o Include the dataset size, i.e., number of images/cases.
- Related work:
o Bilal et al., (2023) PMID: 37890902 may be another relevant work to cite.
- Methods:
o Additional dataset details would be informative, e.g. from what time period/dates were the cases acquired? And how many separate cases/patients were the 2000 images acquired from?
o More details on the scanning step e.g. scanner model used and image acquisition settings.
o Was the expert pathologist who classified the slides blinded to the original diagnosis?
- Figures/Tables:
o Figure 2: more detail in this figure would be useful, e.g. the number of cases, images, and patches at each stage. It is also unclear why there seem to be two labelling steps?
o Table 3: there should be consistency in number of decimal places displayed. It is also unclear why precision/recall are not reported for some models (an explanation in the legend would be helpful).
o Figure 5: the axes are cut off in the figure.
o Figure 6 and 7 are identical (presumably this should be just one single figure, not two).
o It would be preferable for viewing/comparison if Figures 8 and 9 were combined into one figure.
o All figure/table legends: more descriptive detail in the legends would aid interpretation.

There are several statements in the manuscript which seem to unequivocally suggest that utilising AI for diagnostic tasks in histopathology is proven to be an effective approach (examples below). However, it should be acknowledged that while there is strong potential in this area, from a clinical perspective, this field is still in its infancy, and it has not been fully proven that DL approaches can be effective for diagnosis in a routine clinical setting. The language should be toned down to take this into account:
- Abstract:
o “The integration of deep learning architectures in digital pathology proved to be an efficient and effective tool for assisting pathologists in diagnosing cancerous tissues”.
- Introduction:
o “The digital pathology field has had numerous breakthroughs over the years with the advancements in Artificial Intelligence (AI) and Deep Learning (DL) algorithms, where these histopathological images are utilized by such algorithms for an effective diagnosis [6]."
o "The recent innovations in DL architectures presented significant enhancements in CRC diagnosis by making it an efficient and reliable tool for pathologists to diagnose the cases effectively [7]."
o "DL architectures have proven to be effective in extracting complex and high-level features from images, making them suitable and reliable in diagnosing cancerous tissues [8]."

Minor clarification:
- “…where the tumors undergo several phases before placing it in microscopic slides”. It is not clear what is meant by ‘phases’ in this sentence – more specific language would improve the understanding here.

Experimental design

Missing methodological details noted above.

Validity of the findings

The authors compare the performance of their model on the test set, with the performance of other published methodologies (Table 3). However, it should be clearly acknowledged that this is not an entirely fair comparison, as the current model was trained on data from this same location, whereas the other models were not.

Some discussion and/or description of the samples that were misclassified as benign would be beneficial to understand where there are areas for improvement in the model.

---

## Round 0.2 · Minor Revisions

· Academic Editor

Minor Revisions

**Language Note:** When you prepare your next revision, please either (i) have a colleague who is proficient in English and familiar with the subject matter review your manuscript, or (ii) contact a professional editing service to review your manuscript. PeerJ can provide language editing services - you can contact us at [email protected] for pricing (be sure to provide your manuscript number and title). – PeerJ Staff

·

Basic reporting

-

Experimental design

Evaluation and methods as described by the authors seem adequate. Discussion is somewhat suboptimal. This may still be acceptable. Citations are fine.

Validity of the findings

Small sample as compared to some of the other models, such as UNI, CONCH, and Virchows, so the authors would require adjusting the claims.

Reviewer 2 ·

Basic reporting

The majority of my comments have been addressed in the revised manuscript submitted by the authors. However, several of the clarifications relating to the figures/tables seem to have been overlooked, e.g.:

- Figure 2 – it remains unclear why there are two labelling steps – the distinction between what is meant by ‘labelling’ in step 2 and ‘labelling’ in step 4 should be explained in the legend or within the figure, or a different term other than ‘labelling’ should be used for one of these steps to make the distinction.

- Table 3 – There should be consistency in the number of decimal places displayed. It is also unclear why precision/recall are not reported for some models (an explanation in the legend or text would be helpful).

- Figures 6 and 7 still appear to show identical images.

Experimental design

-

Validity of the findings

The text addition “It is important to note that the comparison presented in Table 3 may not be entirely equitable…” seems to have been repeated in the revised manuscript (pp. 17-18).

---

## Round 0.3 · Minor Revisions

· Academic Editor

Minor Revisions

Please follow the requests, criticisms, and suggestions of the reviewers assiduously.

·

Basic reporting

No further issues.

Experimental design

Data preprocessing is lightly touched.
Evaluation methods, assessment metrics, and model selection methods are adequately described.

Validity of the findings

Experiments and evaluations performed adequately in my view, though no assessment of the impact or novelty was available.

Conclusions identify unresolved questions/limitations/ future directions to a fair extent.

Reviewer 2 ·

Basic reporting

The authors have addressed my outstanding comments.

Experimental design

-

Validity of the findings

-

---

## Round 0.4 · accepted · Accept

· Academic Editor

Accept

CRC diagnosis will be critical, and your DL contribution is highly appreciated.

·

Basic reporting

Good

Experimental design

OK

Validity of the findings

Just fine

Additional comments

None

Reviewer 2 ·

Basic reporting

-

Experimental design

-

Validity of the findings

-